# Hypoxic-Ischemic Brain Injury in ECMO: Pathophysiology, Neuromonitoring, and Therapeutic Opportunities

**DOI:** 10.3390/cells12111546

**Published:** 2023-06-05

**Authors:** Shivalika Khanduja, Jiah Kim, Jin Kook Kang, Cheng-Yuan Feng, Melissa Ann Vogelsong, Romergryko G. Geocadin, Glenn Whitman, Sung-Min Cho

**Affiliations:** 1Division of Cardiac Surgery, Department of Surgery, Johns Hopkins University School of Medicine, Baltimore, MD 21287, USA; skhandu2@jhmi.edu (S.K.); jkang71@jh.edu (J.K.K.); gwhitman@jhmi.edu (G.W.); 2Department of Neurology, Johns Hopkins University School of Medicine, Baltimore, MD 21205, USA; jkim684@jh.edu (J.K.); cfeng17@jh.edu (C.-Y.F.); 3Department of Anesthesiology, Perioperative & Pain Medicine, Stanford University School of Medicine, Stanford, CA 94305, USA; mvoge@stanford.edu; 4Divisions of Neurosciences Critical Care, Departments of Neurology, Surgery, Anesthesiology and Critical Care Medicine and Neurosurgery, Johns Hopkins University School of Medicine, Baltimore, MD 21205, USA; rgeocad1@jhmi.edu

**Keywords:** hypoxia-ischemia brain injury, extracorporeal membrane oxygenation, neurological complication, neuromonitoring, outcome

## Abstract

Extracorporeal membrane oxygenation (ECMO), in conjunction with its life-saving benefits, carries a significant risk of acute brain injury (ABI). Hypoxic-ischemic brain injury (HIBI) is one of the most common types of ABI in ECMO patients. Various risk factors, such as history of hypertension, high day 1 lactate level, low pH, cannulation technique, large peri-cannulation PaCO_2_ drop (∆PaCO_2_), and early low pulse pressure, have been associated with the development of HIBI in ECMO patients. The pathogenic mechanisms of HIBI in ECMO are complex and multifactorial, attributing to the underlying pathology requiring initiation of ECMO and the risk of HIBI associated with ECMO itself. HIBI is likely to occur in the peri-cannulation or peri-decannulation time secondary to underlying refractory cardiopulmonary failure before or after ECMO. Current therapeutics target pathological mechanisms, cerebral hypoxia and ischemia, by employing targeted temperature management in the case of extracorporeal cardiopulmonary resuscitation (eCPR), and optimizing cerebral O_2_ saturations and cerebral perfusion. This review describes the pathophysiology, neuromonitoring, and therapeutic techniques to improve neurological outcomes in ECMO patients in order to prevent and minimize the morbidity of HIBI. Further studies aimed at standardizing the most relevant neuromonitoring techniques, optimizing cerebral perfusion, and minimizing the severity of HIBI once it occurs will improve long-term neurological outcomes in ECMO patients.

## 1. Introduction

The use of extracorporeal membrane oxygenation (ECMO) as a supportive/rescue therapy for refractory cardiac or respiratory failure has increased dramatically over the last decade [1]. ECMO has shown survival benefits but carries a substantial risk of acute brain injury (ABI), which can lead to increased morbidity and mortality [2]. 

The brain is susceptible to injury in response to decreased blood supply due to its high metabolic demand. The term “ABI” includes ischemic stroke, hypoxic-ischemic brain injury (HIBI), intracerebral hemorrhage, subdural hemorrhage, subarachnoid hemorrhage, and brain death. Targeted clinical [3,4,5] and pathological studies [6,7,8] report a significantly higher prevalence of ABI in ECMO patients compared to registry studies, e.g., those supported by the Extracorporeal Life Support Organization (ELSO). Even non-registry studies underestimate the frequency of ABI due to a lack of standardized neuromonitoring and the challenges of imaging patients on ECMO [8,9,10,11]. Notably, ABI in the setting of ECMO approximately doubles an already-high mortality risk [8,9]. In a retrospective analysis of brain autopsy in ECMO patients, HIBI was reported as the most common type of ABI in ECMO. HIBI presents as global hypoxic-ischemic encephalopathy associated with widespread brain injury, which is noted more in the cerebral cortices (82%) than in the subcortical and infratentorial areas (i.e., cerebellum 55%, brainstem 36%, and basal ganglia 18%) [12]. In ECMO-supported patients, HIBI presents as significant neurological impairment, ranging in severity from mild cognitive deficits, persistent vegetative state to coma [13,14]. 

Numerous studies have reported ABI in ECMO patients but specific data pertaing to HIBI is sparse. Gaining a better understanding of the prevalence, mechanisms, and neuromonitoring of HIBI in ECMO might significantly improve our management of these critically ill individuals. In this narrative review, we aim to synthesize the available literature on HIBI in ECMO patients, focusing on the prevalence, risk factors, underlying pathophysiology, outcomes, neuromonitoring strategies, therapeutic techniques, and future directions. Our goal is to contribute to the development of effective clinical practices that decrease the burden of HIBI in ECMO patients. 

## 2. Clinical Evidence of HIBI in ECMO

### 2.1. Epidemiology 

ABI is one of the most common complications of ECMO, though prevalence varies significantly by both diagnostic methodology and subpopulation [9,15]. In a meta-analysis by Shoskes et al., ABI was reported in 16% of patients on ECMO, of which half were HIBI [16]. When a standardized neuromonitoring protocol was used in a single retrospective analysis, ABI was diagnosed in 33% of ECMO patients, of which 35% were characterized by HIBI [17]. Through direct tissue examination, an autopsy study of 25 ECMO patients by Cho et al. identified ABI in 68% of patients, two-thirds of which were characterized by HIBI [12]. Furthermore, in a detailed analysis comparing the risks with different cannulation strategies, more cases of ABI and HIBI were reported with venoarterial ECMO (VA-ECMO) compared to venovenous ECMO (VV-ECMO), for ABI: 19% vs. 10%; (*p* = 0.002) and for HIBI: 13% vs. 1%; (*p* < 0.001) [16]. Meanwhile, in a meta-analysis of extracorporeal cardiopulmonary resuscitation (eCPR), Migdady et al. reported ABI in 27% of patients, of which 23% were constituted by HIBI [2]. Although there are inconsistencies in the definition and reporting methodology, the prevalence of ABI and the portion of those patients suffering from HIBI appears to be substantial. 

### 2.2. Timing and Etiology of HIBI

In clinical terms, HIBI can be attributed to two primary factors: A diffuse perfusion deficit and/or a global hypoxic insult to the brain [18]. The timing of HIBI varies, occurring (1) before ECMO cannulation; (2) during ECMO support; or (3) during the decannulation period [8,11]. 

For VA-ECMO patients, the inciting event surrounding cannulation is derived from poor oxygen delivery and includes cardiac arrest or severe shock, such as that due to post-cardiotomy shock, cardiomyopathy, myocardial infarction, cardiac arrhythmia, fulminant myocarditis, or massive pulmonary embolism [19,20,21]. On the other hand, VV-ECMO can cause HIBI as a result of hypoxic perfusion, not hypoperfusion, and often is associated with acute respiratory distress syndrome (ARDS) and/or progression or exacerbation of underlying chronic pulmonary disease. The underlying causes of ARDS include bacterial, viral or fungal pneumonia, aspiration pneumonitis, or pancreatitis [22]. 

While patients are on VA-ECMO support, lack of pulsatile blood flow in and of itself can be associated with ABI. In a recent study [23], HIBI occurred in 34% of patients with ABI and low pulse pressure was an independent risk factor. The mechanism is thought to be endothelial dysfunction, decreased cerebral oxygen consumption with disruption of cerebral autoregulation [24]. Differential hypoxia (also known as Harlequin or North-South syndrome) is another etiology for HIBI unique to peripherally cannulated VA-ECMO patients with hypoxic respiratory failure coupled with cardiogenic shock (Figure 1). It has been reported in 8.8% of all VA-ECMO patients and represents an avoidable cause of HIBI [25]. In the context of hypoxic respiratory failure, the ejection of deoxygenated blood from left ventricle (by the ventricle itself or a percutaneous left ventricular assist device) can lead to hypoxic cerebral perfusion and a global brain injury. The therapeutic options to manage this phenomenon include: Optimization of oxygen delivery to the native lung (e.g., through ventilator manipulation), changing to a central cannulation via the ascending aorta or the right subclavian artery, veno-arteriovenous cannulation (VAV)-ECMO, or decannulation versus de-escalation to VV-ECMO if there is sufficient cardiac recovery [26]. 

For patients on VV-ECMO support, recirculation and low ECMO-flow-to-cardiac-output ratio are two common underlying etiologies for persistent peripheral systemic hypoxemia [27], which can result in HIBI. Normally, after passing through the oxygenator, oxygenated blood is directed back into the body at a level at or near the right atrium. From there, it passes through the tricuspid valve and enters the pulmonary circulation. Recirculation occurs when the oxygenated ECMO outflow is drained by the ECMO inflow cannula, preventing it from effectively returning to the right ventricle, leading to futile recirculation of ECMO flow. The sine qua non of recirculation is an abnormally high venous drainage O_2_ saturation [28]. If recirculation is not a concern, appropriate systemic Hb-O_2_ saturation is a function of the ratio of ECMO-flow-to-cardiac-output flow. The patient’s native output is often supranormal, while there is a limit to the maximum ECMO flows that can be achieved. An excess in native cardiac output relative to ECMO flow will thus result in blood bypassing the ECMO circuit, effectively creating shunt and worsening hypoxemia.

In both VV- and VA-ECMO patients, issues such as significant blood loss, air entrainment, low pump flow, or oxygenator thrombus can result in ECMO failure, leading to systemic desaturation or hypoperfusion and increasing the risk of HIBI [27].

HIBI can also occur in the peri-decannulation period. The decannulation process itself can cause physiological stress and instability. Risk of end-organ hypoperfusion with neurological complications must be guarded against when weaning from ECMO [29]. 

### 2.3. Risk Factors 

The risk factors for ABI vary due to inherent practice differences across ECMO centers including the devices used, anticoagulation strategies, and individual patient differences [30,31]. There are seven studies that have explored risk factors associated with ABI in ECMO, but only four studies specifically address the risk factors for ABI due to HIBI [16,17,23] (Table 1).

In patients undergoing VA-ECMO, a history of hypertension is associated with ABI and HIBI, possibly the result of a rightward shift of the cerebral autoregulation curve leading to hypoperfusion at blood pressures normally counteracted by vasodilatory mechanisms. In this setting, higher levels of blood pressure may be required to maintain adequate cerebral perfusion [6]. Furthermore, high lactate levels on day 1 of ECMO and low pH (acidosis) during this peri-ECMO cannulation period—both of which are markers of systemic hypoperfusion—were other risk factors seen [12]. Additional risk factors linked to ABI that might potentially be linked to HIBI are an early low pulse pressure (<20%) or a loss of pulsatility, which is associated with endothelial dysfunction, decreased local oxygen consumption, and disruption of cerebral autoregulation, which increases the risk of ABI and HIBI [23]. A high pre-cannulation PaCO_2_ and large peri-cannulation PaCO_2_ drop (∆PaCO_2_) were also associated with an increased ABI risk [17]. Acute decreases in PaCO_2_ induce vasoconstriction of cerebral vessels, which decreases cerebral blood flow (CBF) and increases the risk of ischemia [32,33,34]. Hypocapnia also increases neuronal excitability and cerebral metabolic demand, exacerbated by any decreased CBF, increasing the risk of ABI [32,35,36]. In a retrospective analysis by Shou et al., severe hyperoxia (≥300 mm Hg) following ECMO initiation was associated with ABI and mortality [37]. Therefore, optimization of ABG parameters during ECMO has the potential to decrease ABI and improve survival by increasing carbon dioxide clearance. 

In patients undergoing VV-ECMO, risk factors for ABI and HIBI include lower pH, hypoxemia during the peri-cannulation period, and markers of coagulation disturbances (D-dimer, fibrinogen) [38]. 

### 2.4. Outcome

No studies have specifically investigated the neurological outcome of HIBI in ECMO. However, there are some studies that have identified factors related to neurological prognosis of patients undergoing ECMO cannulation for cardiac or respiratory arrest, which are known to be the main causes of HIBI. A meta-analysis by Migdady et al. reported that 24% of patients with eCPR achieved a good neurological outcome. This meta-analysis also showed that 27% of patients had at least one neurological complication, of which 85% were constituted by HIBI [2]. In a retrospective analysis, Chambrun et al. reported a 1-year survival rate of 27% among patients treated with VA-ECMO after refractory shock post-cardiac arrest, and all of them achieved a favorable neurological outcome [39]. Low hemoglobin or high serum lactic acid levels before ECMO, and prolonged interval from cardiac arrest to ECMO initiation were associated with poor neurological outcomes after successful eCPR [40]. Poor neurological prognosis is perceived as an important cause of withdrawal of life sustaining therapy (WLST) within 72 h after ECMO cannulation. In a study by Carlson et al., early WLST was observed in more than half of the eCPR patients. Early WLST prevents a complete understanding of prevalence, timing, and mechanism of HIBI and consequent neurological complications in ECMO patients [41]. Further clinical studies are warranted to define the relationship of HIBI to the outcome in ECMO patients. 

## 3. Preclinical Models of HIBI in ECMO

A limited number of preclinical models that investigate HIBI in ECMO exist (Table 2). While there are 21 preclinical studies that investigated neurological outcomes in eCPR, we identified three studies that specifically examined HIBI in ECMO. Foerster et al. investigated the effect of anticoagulation during eCPR on neurological outcomes. In this study, twelve pigs (six with anticoagulation, six without anticoagulation) were placed on VA-ECMO after 15 min of cardiac arrest. Neurological outcomes were assessed using the neurological deficit score (NDS), electroencephalography (EEG), magnetic resonance imaging (MRI), and histological examination. Brain histology after 7 days of cardiac arrest revealed moderate hypoxic-ischemic damage in both groups, as evidenced by the appearance of dark neurons and eosinophilic neurons in hippocampus, cerebellum, and frontal lobe; however, it is still in the reversible state of brain ischemia [42]. Although this study demonstrated that eCPR intervention after 15 min of cardiac arrest was associated with a moderate degree of HIBI based on histological findings and that anticoagulation did not mitigate HIBI, the study does not allow for distinction between the impact of eCPR versus that of the arrest and associated global hypoxic injury itself on HIBI.

In a study of 14 pigs performed by Putzer et al., low-flow VA-ECMO was initiated after 8 min of cardiac arrest. After 10 min, adrenaline was continuously infused to achieve mean arterial pressure (MAP) of 40 or 60. Low-flow ECMO prior to adrenaline administration resulted in an inadequate MAP and cerebral perfusion pressure (CPP). This insufficiency contributed to HIBI, as indicated by increased levels of cerebral microdialysis markers (lactate, pyruvate, and lactate to pyruvate ratio). Adrenaline infusion increased MAP, CPP, regional CBF, and cerebral oxygen supply, thereby mitigating hypoxic-ischemic brain damage during low-flow eCPR [43]. This study showed that after an 8-min-long cardiac arrest, HIBI could be mitigated by enhancing MAP and CPP through the use of adrenaline during VA-ECMO therapy. More recently, Rozencwajg et al. studied the effect of VA-ECMO flow on brain injury in six sheep. After inducing severe cardiorespiratory failure, the animals were randomized into two groups—three with low-flow at 2.5 L/min and three with high-flow at 4.5 L/min. Neurological outcomes were thoroughly evaluated by examining brain hemodynamics, oxygenation, metabolism, and histology after 6 h. Although both groups exhibited a similar pattern of HIBI on histology, the low-flow group had significantly more severe histological brain injury, characterized by neuronal shrinkage, congestion, and perivascular edema. An increase in cerebral metabolites suggestive of anaerobic metabolism, such as lactate, pyruvate, and the lactate-to-pyruvate ratio further corroborated the presence of HIBI in the low-flow group. Inadequate oxygenation, as observed through continuous brain tissue oxygen (PbtO_2_) and near-infrared spectroscopy (NIRS), was consistent with the hypoxia and hypoperfusion seen in HIBI in the low-flow group [44]. 

In summary, preclinical studies investigating HIBI in ECMO appear to demonstrate mitigation of HIBI resulting from the institution of ECMO with adequate flow and blood pressure. Whether ECMO in and of itself contributes to HIBI is unanswered, although its benefits clearly outweigh its drawbacks when examining oxygen delivery and the overall prevention of an ischemic/hypoxic brain injury. The determination of optimal targets for important physiological variables is lacking, such as pH, O_2_, CO_2_, MAP, and temperature. There is a pressing need for more preclinical research that employs multimodal neuromonitoring strategies to evaluate neurological outcomes and improve our understanding of HIBI in the setting of ECMO rescue and support.

## 4. Pathophysiology

The pathophysiology of HIBI is complex as it depends on the underlying pathology necessitating the use of ECMO along with the risk of HIBI associated with ECMO support itself. 

At the molecular cellular level, sudden cessation of CBF initiates the cascade of neuronal ischemia and subsequent cell death within minutes [45]. Energy production is impaired and adenosine triphosphate (ATP) stores are quickly depleted. The ATP-dependent Na^+^/K^+^ ATPase can no longer maintain electrolyte homeostasis, and thus sodium and water begin to accumulate inside the cell leading to cytotoxic edema [46]. Oxidative phosphorylation shifts to anaerobic glycolysis [47] with a resultant accumulation of lactate, leading to intracellular and extracellular acidosis and further impairing the cell function. Neuronal ischemia activates the N-methyl-D-aspartate (NMDA) receptor, which causes an influx of calcium ions. Together with the release of excitotoxic neurotransmitter glutamate and formation of reactive oxygen species, numerous enzymes are released and activated, such as lipases, proteases, and nucleases, which accelerate cell death [48,49]. Three types of neuronal death have been described [18]: Necrosis, apoptosis, and autophagocytosis. Both calcium influx and NMDA receptor activation contribute, and in combination with the production of nitric oxide and free radicals, apoptosis and autophagocytosis occur (Figure 2). 

At the functional cellular level, HIBI causes mitochondrial dysfunction which impairs ATP production and increases free radical synthesis. Mitochondrial dysfunction can dismantle cell structures (for example, demyelination and cytoskeletal damage), impair the blood–brain barrier, and halt synaptic transmission. HIBI can upregulate (e.g., Bcl-2, HSP) or downregulate various genes to protect against or facilitate cell damage caused by mitochondrial dysfunction [18,50].

At the tissue level, the ECMO circuit converts the blood into a prothrombotic and proinflammatory state (Figure 1). Together with the pre-ECMO low-flow state, and the subsequent continuous non-pulsatile flow mode, all these can result in endothelial injury, microcirculation disruption, and then eventually cerebral autoregulation impairment [51]. In this context, CBF becomes extremely sensitive to systemic blood pressure and vulnerable to blood pressure changes as a result of loss of cerebral autoregulation. The patient with pre-ECMO essential hypertension is especially at risk given that the cerebral autoregulation curve has already been shifted to the right [6,12]. 

Evidence suggests that a spectrum of epigenetic processes play a crucial role in the pathophysiology of HIBI [52]. The epigenetic mechanisms that have a role in regulation of vascular and neuronal regeneration after HIBI include DNA methylation, histone deacetylase, and microRNAs (miRNAs). 

DNA methylation status changes dramatically after HIBI. It could present as an increased or decreased methylation of different genes. HIBI is associated with an increased expression of the genes related to angiogenesis and apoptosis (Casp1, Casp9, Casp8ap2, vascular endothelial growth factor a (VEGFa), VEGFc, Epor, Epo, Hif 1α, and Hif 3α). 

The outcome of cerebral ischemia is greatly influenced by the acetylation status of histones. The creation of an ischemia-resistant state in neurons requires histone acetylation and cAMP-response element binding protein (CREB)-binding protein (CBP). The neurological outcomes are highly correlated with the expression levels of histone acetylation and CBP.

miRNAs in the brain and blood can be used as biomarkers for cerebral ischemia. Animal studies have shown that the expression of miRNA in the hippocampus of rats is altered after global ischemic insult. miRNAs regulate the normal physiological activity in conjunction with the response to ischemia–reperfusion injury of the hippocampus [53]. The role of miRNA in the pathogenesis of HIBI is significant, which can be explained by changes in the expression of miRNA (mir-182, mir-200b, and mir-429) after HIBI in rats [54]. The expression of mature miR-139-5p is decreased after HIBI, which leads to increased expression of a newly identified protein, HGTD-P, and consequently promotes neuronal apoptosis in neonatal rats after HIBI [55]. miR-210 prevents oxygen–glucose deprivation-induced apoptosis and protects cells against HIBI [56].

Research into the roles of epigenetic mechanisms in cerebral ischemia is rapidly growing. However, our understanding of the epigenetic regulations for HIBI is still in its infancy. The epigenetic strategies targeting gene expression can be used for the treatment of HIBI, such as the inhibition of DNA methyltransferase activities, histone deacetylase enzyme, and miRNAs. Neuroprotective agents targeting these pathways can modulate neural cell regeneration and promote brain repair and enhance functional recovery after HIBI. A better understanding of how epigenetics influences the process and progress of cerebral ischemia will pave the way for discovering more sensitive and specific biomarkers as well as new targets and therapeutics for HIBI, and will improve the neurological outcomes. 

## 5. Neuromonitoring

### 5.1. Serial Neurological Examination 

The serial neurological examination is essential in evaluating acute neurological changes. While sedation minimization is frequently a goal, sedation remains as a necessity for many patients supported by ECMO; therefore, a complete neurological examination is not always possible. Noninvasive neuromonitoring may be crucial in these patients with impaired consciousness as it has the ability to rapidly detect ABI [57,58]. In a study conducted by Ong et al., an increased detection of ABI (from 23% to 33%) and improved neurological outcomes at discharge (from 30% to 54%) were observed with the introduction of standardized neuromonitoring for ECMO patients [59].

### 5.2. EEG

EEG is a noninvasive tool that measures cortical electrical activity with spatial and temporal resolution and is sensitive to changes in brain structure and function [57]. EEG monitoring can be intermittent or continuous to evaluate changes in generalized or focal background. In a meta-analysis study by Perera et al., EEG was highly specific for disability and death when status epilepticus, burst suppression, or electrocerebral silence was reported in comatose patients with HIBI secondary to cardiac arrest [60]. Additionally, continuous EEG (cEEG) indicating moderate encephalopathy (diffuse slowing with reactivity/variability) and seizure or generalized periodic discharges are associated with a poor outcome in cardiac arrest [61]. Meanwhile, a recent study by Hwang et al. in comatose ECMO patients showed no epileptiform discharges or seizures, even though the cEEG was performed in the absence of confounding sedating medication. The study also found that intact reactivity presents state changes, and fair/good variability may be associated with survival at hospital discharge. This study implies that these features are a better approach than relying on the presence of “highly malignant” patterns for neurological prognosis [62,63]. According to “The American Clinical Neurophysiology Society Consensus Statement on EEG (cEEG) in Critically Ill Adults and Children”, patients on ECMO requiring sedation and paralysis are designated as a high-risk group that should be monitored by cEEG [64]. The widespread use of cEEG in comatose ECMO patients has the potential to be beneficial in assessing the degree of brain injury and monitoring seizures and brain function, as well as in predicting neurological outcomes [57].

### 5.3. Cerebral NIRS 

Cerebral NIRS noninvasively measures regional oxygen saturation (rSO_2_) by determining the relative concentrations of oxygenated and deoxygenated hemoglobin in the cerebral circulation [65]. Since HIBI is caused by decreased blood flow and oxygen to the brain, NIRS measurement of rSO_2_ can be used as a diagnostic tool. Several studies [66,67,68] have shown that NIRS is capable of detecting ABI in ECMO patients. For example, a significant (>25%) drop from baseline, as well as frequency, duration, and burden of desaturations noted on NIRS have all been related to ABI on ECMO. However, lack of standardization of the monitors and variable sensitivity under differing patient conditions, at least at the moment, makes the readings non-specific. Therefore, thresholds for intervention are unclear [69]. With no specific NIRS data on ECMO-associated HIBI, more studies are warranted to define the utility of NIRS in this group of patients. 

### 5.4. SSEP

Somatosensory evoked potentials (SSEPs) are a neurophysiological monitoring tool that apply mild electrical stimulation in the peripheral nerve and measure the electrical responses along the somatosensory pathway [70]. Negative peak at 20 ms (N20) is the first component recorded from the cortex in median nerve SSEPs [71]. The European Resuscitation Council (ERC) and the European Society of Intensive Care Medicine (ESICM) guidelines include an absence of the N20 SSEP wave among the most robust predictors to be tested at 72 h after return of spontaneous circulation (ROSC) in cardiac arrest patients [72,73]. Moreover, bilaterally absent cortical SSEP response is a very reliable predictor of poor neurological outcome in patients with HIBI. However, SSEP has a high specificity, but a low sensitivity. In other words, even many patients doomed to a poor neurological outcome have a bilaterally present N20 SSEP wave [74]. To address this limitation, several studies measured amplitude changes in the N20 wave at specific time points or combined bilateral absence of the cortical N20 with N20-P25 threshold amplitude results to obtain a better prognostic neurological outcome [75,76]. Furthermore, given the sensitivity of SSEP to a wide range of physiologic factors (blood flow, blood pressure, hematocrit, hypoxia, hypercarbia, increased intracranial pressure (ICP), and commonly used analgosedative agents (e.g., midazolam, propofol, opioids, ketamine) [77,78], SSEP should be cautiously interpreted in ECMO patients. To date, no studies have utilized SSEP alone to predict neurological outcomes in ECMO population. Given the high potential for SSEP to be used as a neurological outcome predictor in ECMO patients, further studies on this topic are required [75,76]. 

### 5.5. SSEP and EEG

SSEPs in combination with EEG findings predict neurological outcomes better than SSEP alone. For example, in cardiac arrest patients, the combination of high N20 amplitude with a benign EEG increased the sensitivity of predicting a favorable neurological outcome from 61% to 91% [79]. A study by Cho et al. analyzed the combined results of SSEP and EEG for predicting neurological outcomes in ECMO patients. ECMO patients with poor neurological outcomes demonstrated a loss of EEG reactivity despite a relatively preserved EEG background and preservation of the SSEP response. This triad of neurophysiological findings may represent a pattern of ECMO-specific brain injury termed “ECMO brain” and suggests that ABI in ECMO patients appears to exhibit a different neurological injury pattern than is seen in other common neurological injuries [80]. 

### 5.6. Transcranial Doppler

Transcranial doppler (TCD) is a traditional neuromonitoring tool that enables the measurement of blood flow velocity in major cerebral arteries. It is commonly utilized in the assessment of cerebrovascular disorders, such as ischemic stroke, subarachnoid hemorrhage, and cerebral vasospasm. However, given its mechanism of action, TCD is not applicable to monitoring HIBI patients, in whom the pathology is mainly related to global hypoxia; rather, TCD is used to monitor ECMO circuit clot, cerebral emboli detection, and cerebral hemodynamics [81,82].

### 5.7. Plasma Biomarkers

Brain injury biomarkers, such as neuron-specific enolase (NSE) and calcium-binding protein B (S100B), have been commonly examined to evaluate brain injury in post-cardiac arrest patients [83]. European guidelines suggest NSE as the preferred biomarker in neuroprognostication [84]. However, hemolysis occurs frequently in ECMO, which can lead to false-positive NSE results [85]. The half-life for NSE and S100B is approximately 24 h and 2 h, respectively. Therefore, it is expected that S100B peaks earlier in the circulation after HIBI than NSE [86]. A systematic review by Wang et al. demonstrated that early S100B or late NSE value functioned as reliable prognostic indicators for post-cardiac arrest patients and the specificity was consistently high regardless of the timing of measurement [87]. Similarly, in children requiring ECMO, a study of six brain injury biomarkers found that NSE and glial fibrillary acidic protein (GFAP) were significantly associated with unfavorable outcomes [88]. These findings suggest that NSE, S100B, and GFAP can be useful in predicting neurological outcomes. A more recent meta-analysis study by Hoiland et al. identified that neurofilament light (Nf-L) had the highest diagnostic accuracy in predicting an unfavorable neurological outcome in patients with HIBI following cardiac arrest. Tau, a marker which reflects axonal injury, had greater diagnostic accuracy compared to NSE or S100B [89]. The prognostic value of biomarkers varies in different meta-analysis studies, emphasizing the need for caution when interpreting results. Furthermore, the complexity of brain injury in patients receiving ECMO is unlikely to be captured by a single biomarker. Biomarkers may help in predicting prognosis as part of a multidisciplinary evaluation that includes imaging and clinical assessment to increase sensitivity [24]. Future studies may need to investigate a panel of biomarkers for each category to better understand the nature of brain injury or combine multiple biomarkers to improve diagnostic accuracy in the ECMO population. Moreover, when using biomarkers for neurological prognostication, the timing of biomarker draws should be considered.

### 5.8. Imaging

#### 5.8.1. Brain CT

Brain computed tomography (CT) is a valuable diagnostic tool for HIBI in ECMO patients. The greatest benefit is that the examination can be performed even while maintaining ECMO. The test can be conducted immediately upon detection of neurological changes, allowing for rapid acquisition and early diagnosis. However, there are certain limitations, including a low resolution, the necessity for transport, radiation exposure, and relative insensitivity to early ischemic injury [66]. When performing brain CT in ECMO patients, several factors need to be considered due to the differences in the pathophysiology of ECMO patients compared to non-ECMO patients. These factors include incompatibility of ECMO circuit components with MRI, logistics of transport, timing of contrast administration, and interpretation of contrast-enhanced CT imaging in the context of altered blood flow patterns [90]. Furthermore, brain CT displays only non-specific signs, such as cerebral edema, sulci effacement, and decreased gray matter (GM)/white matter (WM) differentiation [91]. The regions primarily affected by profound hypoxia-ischemia are the deep gray matter nuclei, cortices, hippocampi, and cerebellum [92]. The pattern or location of the lesion may be helpful in diagnosing HIBI and differentiating it from other diseases with similar patterns. In a study performed by Zotzmann et al., full body CT scans performed within 24 h of ECPR revealed cerebral edema suggestive of HIBI in 26.2% of patients [93]. Overall, brain CT scans have certain limitations, such as low resolution and relative insensitivity to early ischemic injury, but can provide important information for diagnosing HIBI in ECMO patients.

#### 5.8.2. Brain MRI

In the diagnosis of HIBI, brain MRI is highly sensitive and shows different changes depending on the timing of the injury [94,95]. However, the ECMO circuit has components that are not compatible with the magnetic field of the MRI. Consequently, until decannulation and removal of ECMO pump, MRI is not feasible [90]. According to the current ELSO guidelines, neuroimaging in the form of brain CT or MRI should be performed before discharge in neonatal and pediatric ECMO patients for any indication [66]. On the other hand, there are no ELSO guidelines related to brain imaging in adult ECMO patients. Brain imaging is performed as needed when neurological symptoms are present and brain injury is suspected. In the acute phase, diffusion-weighted imaging (DWI) is preferred over T2-weighted imaging for detecting lesions due to the sensitivity of detecting the presence of cytotoxic edema earlier. In the late subacute phase, postanoxic leukoencephalopathy and contrast enhancement could be observed. In the chronic phase, atrophic changes over tissue signal changes are more prevalent [91]. To overcome the limitations of MRI examination, research on low-field portable MRI technology was recently conducted (SAFE MRI ECMO study) [96,97]. A study by Cho et al. on three ECMO patients reported that imaging using low-field MRI can be conducted without significant changes in MAP, ECMO flow, and SpO_2_. Low-field MRI was able to produce high-quality neuroimages superior to brain CT to enable bedside detection of ABI [96]. The evaluation of brain injury to detect HIBI using low-field portable MRI in 19 cardiac arrest patients showed that 12 patients (63.2%) had findings consistent with the findings on conventional MRI report. Low-field MRI was performed without safety events or disrupting intensive care unit equipment setup but had lower signal-to-noise ratio compared to conventional MRI, resulting in degraded image quality [98]. It is anticipated that in the future, early diagnosis of various neurological sequelae of ECMO, including HIBI, will be possible with bedside portable MRI.

## 6. Therapeutic Management

Immediate intervention to reverse hypoxia and ischemia is critical to prevent HIBI, but the exact effects on cerebral perfusion of the circuit and the hemodynamic support achieved by ECMO are unclear. Therefore, at the present time, treatment for ECMO-associated HIBI [51] is similar to that of a typical cardiac arrest and consists of temperature control and management of cerebral edema and elevated ICP.

### 6.1. Temperature Control

Targeted temperature management (TTM), which maintains a set temperature range of 32–36 ℃, is a widely used neuroprotective intervention in patients after cardiac arrest; however, its benefit in the context of eCPR remains unclear. Several studies have evaluated the impact of TTM on survival and neurological outcomes among eCPR patients and none have found a significant benefit [99,100]. Two recent trials on the effect of moderate hypothermia on mortality in patients with cardiogenic shock supported by VA-ECMO [101], and in patients with cardiac arrest [102], showed no improvement in survival; however, neurological outcomes were not assessed in either of the studies. Furthermore, 2022 ERC-ESICM guidelines state that there is insufficient evidence to support benefits/risks of using TTM and actively recommend preventing fever after cardiac arrest [103]. Despite its potential to exacerbate any underlying coagulopathy, TTM should not be dismissed as ineffective in treating HIBI in ECMO. In fact, given the strong basic and preclinical scientific support for temperature control in global ischemia [104] and the high incidence of HIBI and prolonged absent/low cerebral perfusion in VA-ECMO patients, a well-designed multicenter prospective observational cohort study is necessary for better understanding the benefits/risks associated with TTM in VA-ECMO patients as well as eCPR patients. 

### 6.2. Cerebral Edema and Elevated ICP

Several studies [105,106] have been conducted on the premise that ICP may be elevated in HIBI due to brain edema in the setting of an ischemic, hypoxic injury. In a preliminary study by Fergusson et al. on HIBI patients after cardiac arrest, ICP management guided by invasive neuromonitoring was associated with a 6-month favorable neurological outcome [107]. However, the level of ICP in the ECMO population is unknown since invasive ICP monitoring requires anticoagulation to be discontinued, which has its own risks in ECMO patients. Therefore, ICP management in ECMO patients is reserved for patients who clinically exhibit manifestations of transtentorial herniation, such as acute dilated non-reactive pupil to light. ICP control in patients with HIBI undergoing ECMO can be managed using the following commonly employed strategies [51]: (a) Head-of-bed elevation > 30 degrees [84,108], such as with the Reverse Trendelenburg position; (b) Normocapnia with PaCO2 35–45 mmHg is recommended [109], although hyperventilation can be employed as a temporary measure [110], which can easily be achieved by utilizing the gas flow across the oxygenator to maintain PaCO_2_ levels below 30 mmHg; (c) Hyperosmolar therapy with the intermittent administration of hypertonic saline and/or mannitol [110,111]; (d) Control of factors that increase ICP, such as pain, seizure, agitation, and fever; (e) Use of effective analgesics and sedatives, such as propofol, barbiturates [110], and midazolam [106]; and (f) Consideration of neurosurgical intervention, such as decompressive craniectomy, though the benefit of this approach is unknown [51].

## 7. Future Directions

Numerous studies have reported HIBI as a common type of ABI seen in ECMO. As a result, it is important to deepen our understanding of its pathophysiology in these patients, concentrating on the physiological parameters that affect it, over which we have control [50]. The field of ABI and specifically HIBI is wide open to identify and standardize neuromonitoring techniques, new therapies, and understand long-term neurological and functional outcomes [112]. New neuroprotective agents, such as antioxidants, anti-inflammatory agents, and neurotransmitter modulation all show promise in mitigating the severity of HIBI. Furthermore, advances in ECMO technology, such as new circuit membrane materials, alternative cannulation strategies, and increasing pulsatile flow may mitigate HIBI.

Finally, there has been only minimal research on the risk of psychiatric illness or the level of cognitive impairment that determines the quality of life of survivors with HIBI in ECMO-treated patients. The incidence of psychiatric illness including depression, anxiety, insomnia, and PTSD, as well as cognitive recovery after ECMO-related ABI has not been well-described. An approach to caring for these patients that is multidisciplinary will be essential to understand and improve their management.

## 8. Conclusions

ABI results in a two-fold elevation of the already-high mortality risk in the ECMO setting, with HIBI being reported as the most common type of ABI in ECMO patients. The incidence of HIBI is often underestimated under the special circumstances of ECMO, which makes neurological examination unreliable due to sedation and diagnostic imaging difficult, as well as due to the lack of standardized neuromonitoring. The pathogenic mechanisms of HIBI in the setting of ECMO are complex and multifactorial. The considerable heterogeneity within pathophysiology poses additional challenges to research, diagnosis, and the selection of appropriate treatment. This review comprehensively describes the pathophysiology, risk factors, neuromonitoring tools, and therapeutic approaches for HIBI in ECMO patients. The novelty and significance of our study lie in its exclusive focus on HIBI within the ECMO population. We anticipate that this study will enhance our comprehension of HIBI within the ECMO setting, increase HIBI diagnosis rates, and encourage further research into treatment. Nonetheless, many aspects of this patient population remain unclear. Therefore, future research is needed to gain a better understanding of cellular and molecular mechanisms specific to the type of ECMO, standardize the most appropriate neuromonitoring techniques, optimize cerebral perfusion during ECMO, and investigate novel neuroprotective agents and neurotransmitter modulation to mitigate the severity of HIBI.

## Figures and Tables

**Figure 1 cells-12-01546-f001:**
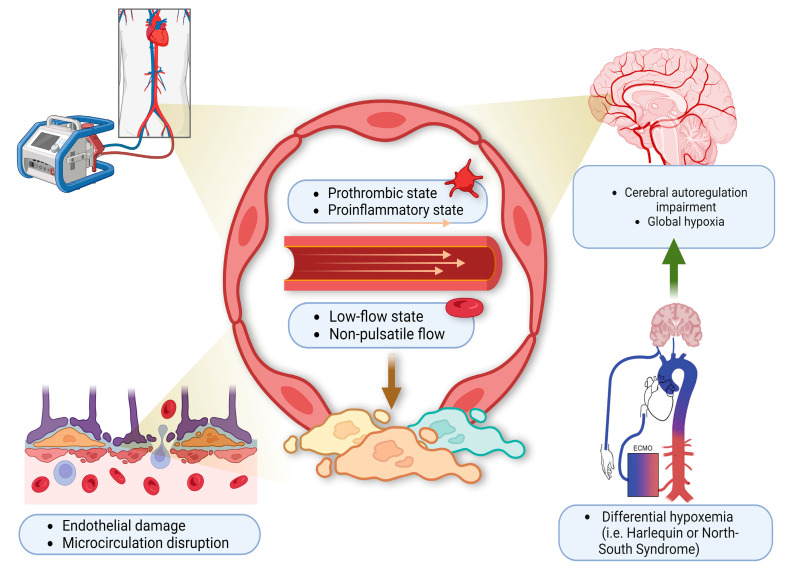
**Harlequin syndrome.** ECMO circuit induces hypercoagulable state due to a low-flow state and non-pulsatile blood flow. This leads to endothelial injury, microcirculation disruption, and differential hypoxemia. Over time, as the left ventricle regains its function, deoxygenated blood is pumped by the heart to the cerebral blood vessels leading to cerebral autoregulation impairment and global hypoxia. This clinically presents as HIBI. This figure is created with the BioRender software, with the assistance of Soorin Chung in finalizing the graphic design.

**Figure 2 cells-12-01546-f002:**
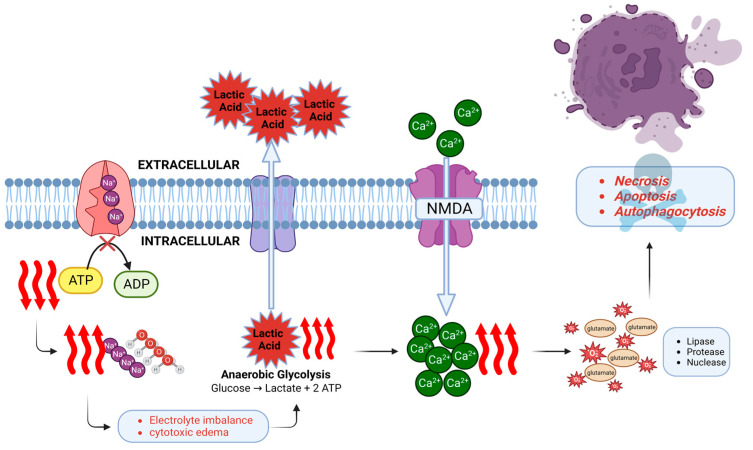
**Pathogenesis of HIBI at the molecular level.** Decreased cerebral perfusion impairs the function of Na^+^/K^+^ ATPase pump in the brain tissue due to decreased ATP production. This generates a hypoxic state which leads to a shift in metabolism from oxidative phosphorylation to anaerobic glycolysis. The resultant acidotic environment due to lactic acid accumulation impairs cell function. Furthermore, ischemia leads to an activation of NMDA receptor and influx of calcium ions. Calcium induces the release of excitotoxic neurotransmitter glutamate, which along with the reactive oxygen species and activation of degrative enzymes leads to neuronal cell death by necrosis, apoptosis, or autophagocytosis. This figure is created with the BioRender software, with the assistance of Soorin Chung in finalizing the graphic design.

**Table 1 cells-12-01546-t001:** Published trials on risk factors associated with HIBI in ECMO patients.

			**Population Characteristics**		
**Author, Year**	**Study Design**	**Sample Size (*n*)**	**Inclusion Criteria**	**Risk Factors**	**Overall (%)**
Cho, 2020 [12]	Retrospective Cohort (Autopsy)	25	ECMO (88% VA-ECMO)	Hypertension history, high day 1 lactate level, low pH level	ABI (68%) HIBI (44%)
Shoskes, 2020 [16]	Systematic Review and Meta-analysis	16,063	VA-ECMO vs. VV-ECMO	Cannulation method (VA-ECMO)	VA-ECMO vs. VV-ECMO: ABI (19% vs. 10%; *p* = 0.002) HIBI (13% vs. 1%; *p* < 0.001)
Shou, 2022 [17]	Retrospective Cohort	129	VA-ECMO	High pre-cannulation PaCO_2_, large peri-cannulation PaCO_2_ drop (ΔPaCO_2_)	ABI (33%) HIBI (12%)
Shou, 2022 [23]	Retrospective Cohort	123	VA-ECMO	Early low pulse pressure (<20 mmHg)	ABI (33%) HIBI (11%)

**Table 2 cells-12-01546-t002:** Preclinical models of HIBI in ECMO.

Author, Year	Objective	Animal	Size	ECMO Type	ECMO Duration	Intervention	HIBI Findings
Foerster, 2013 [42]	To investigate the effect of anticoagulation during ECPR	Pig	12	ECPR	60 min	ECPR (80–100 mL/kg/min) started after 15 min of cardiac arrest No anticoagulation before ECPR reperfusion (*n =* 6) Heparinized saline solution flush (*n =* 3) Anticoagulant-coated cannulae and normal saline solution flush (*n =* 3)	No difference between the two groups Brain histology after 7 days of cardiac arrest in both groups showed dark neurons and eosinophilic neurons in hippocampus, cerebellum, and frontal lobe
Putzer, 2021 [43]	To investigate options for the use of ECPR without preceding systemic heparinization after cardiac arrest and the effect on survival and neurological outcome	Pig	14	ECPR	10 min	ECPR (30 mL/kg/min) started after 8 min of cardiac arrest Adrenaline infusion for goal MAP 40 (*n =* 7) vs. MAP 60 (*n =* 7)	Microdialysis markers (lactate, pyruvate, and lactate to pyruvate ratio) significantly decreased in MAP 60 group with adrenaline infusion
Rozencwajg, 2023 [44]	To study the impact of the ECMO flow on brain injury	Sheep	6	VA-ECMO	300 min	Low-flow at 2.5 L/min (*n =* 3) High-flow at 4.5 L/min (*n =* 3)	Neuronal shrinkage, congestion, and perivascular edema were higher in the low-flow group PbtO2 levels were lower in the low-flow group NIRS was lower in the low-flow group

## Data Availability

Not applicable.

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
