# Peer review of "Hypoxic-Ischemic Brain Injury in ECMO: Pathophysiology, Neuromonitoring, and Therapeutic Opportunities"

_cells, 2023, doi:10.3390/cells12111546_

Round 1

Reviewer 1 Report

The manuscript "Hypoxic-Ischemic Brain Injury in ECMO: Pathophysiology, Neuromonitoring, and Therapeutic Opportunities" describes the pathophysiology, neuromonitoring, and therapeutic techniques to improve neurological outcomes in extracorporeal membrane oxygenation (ECMO) in patients so as to prevent and minimize the morbidity of hypoxic-ischemic brain injury (HIBI).
The relevance of the work and its novelty are beyond doubt.
ACUTE BRAIN INJURY is one of the common causes of disability and death in the population. The existing huge heterogeneity within the pathology creates additional difficulties in the research, complexities in the diagnosis and selection of the correct treatment. HIBI is incredibly complex process, and therefore any discussion of the underlying mechanisms requires a good basis. Therefore, future research is needed to understand more specific neuronal and molecular mechanisms of brain injury.
However, the initial PubMed search (https://pubmed.ncbi.nlm.nih.gov/), including terms: hypoxic-Ischemic Brain Injury, ECMO releases about 60 articles.
A separate advantage of the review would be the presence of useful summary tables that provide a comparative analysis.
Despite the positive impression of the work, there are a number of minor comments and issues that could improve the manuscript:

Recent reviews have told of the emerging role of epigenetics in the pathophysiology of CNS injury, stroke, and spinal cord injury. The importance of epigenetic mechanisms in neuroplasticity, learning, and memory is also explained. Evidence from recent studies supports the involvement of epigenetic mechanisms such as DNA methylation, posttranslational modification of chromatin, and regulation of miRNA gene expression in the post-traumatic brain.  E.g., Bertogliat, M. J., Morris-Blanco, K. C., & Vemuganti, R. (2020). Epigenetic mechanisms of neurodegenerative diseases and acute brain injury. Neurochemistry international, 133, 104642. https://doi.org/10.1016/j.neuint.2019.104642
These studies reinforce our current understanding of the structural and functional changes induced by TBI and may provide innovative approaches to TBI recovery and therapeutic intervention.
The role of epigenetic changes in this condition is still little studied, but this is a promising area of research. As a recommendation, it may be worthwhile to include in this review recent research on epigenetic mechanisms in brain ischemia/hypoxia.
In the section describing pathogenesis, it would be a good idea to add timing of events occurring at the molecular-cellular level (Timeline).

The authors were able to make a significant contribution to the development of effective clinical practices that reduce the burden of HIBI in patients with ECMO.

I recommend this manuscript for publication, after implementing the suggested improvements.

Author Response

Reviewer 1

The manuscript "Hypoxic-Ischemic Brain Injury in ECMO: Pathophysiology, Neuromonitoring, and Therapeutic Opportunities" describes the pathophysiology, neuromonitoring, and therapeutic techniques to improve neurological outcomes in extracorporeal membrane oxygenation (ECMO) in patients so as to prevent and minimize the morbidity of hypoxic-ischemic brain injury (HIBI).
The relevance of the work and its novelty are beyond doubt.
ACUTE BRAIN INJURY is one of the common causes of disability and death in the population. The existing huge heterogeneity within the pathology creates additional difficulties in the research, complexities in the diagnosis and selection of the correct treatment. HIBI is incredibly complex process, and therefore any discussion of the underlying mechanisms requires a good basis. Therefore, future research is needed to understand more specific neuronal and molecular mechanisms of brain injury.
However, the initial PubMed search (https://pubmed.ncbi.nlm.nih.gov/), including terms: hypoxic-Ischemic Brain Injury, ECMO releases about 60 articles.
A separate advantage of the review would be the presence of useful summary tables that provide a comparative analysis.
Despite the positive impression of the work, there are a number of minor comments and issues that could improve the manuscript:

Thank you for the suggestion and feedback. Appreciate it greatly.

Recent reviews have told of the emerging role of epigenetics in the pathophysiology of CNS injury, stroke, and spinal cord injury. The importance of epigenetic mechanisms in neuroplasticity, learning, and memory is also explained. Evidence from recent studies supports the involvement of epigenetic mechanisms such as DNA methylation, posttranslational modification of chromatin, and regulation of miRNA gene expression in the post-traumatic brain.  E.g., Bertogliat, M. J., Morris-Blanco, K. C., & Vemuganti, R. (2020). Epigenetic mechanisms of neurodegenerative diseases and acute brain injury. Neurochemistry international, 133, 104642. https://doi.org/10.1016/j.neuint.2019.104642
These studies reinforce our current understanding of the structural and functional changes induced by TBI and may provide innovative approaches to TBI recovery and therapeutic intervention.
The role of epigenetic changes in this condition is still little studied, but this is a promising area of research. As a recommendation, it may be worthwhile to include in this review recent research on epigenetic mechanisms in brain ischemia/hypoxia.

Thank you for this comment. We agree with you. We now have incorporated the details pertaining to the epigenetics and hypoxic-ischemic brain injury on page 9.

  • Evidence suggests that a spectrum of epigenetic processes play a crucial role in the pathophysiology of HIBI. The epigenetic mechanisms that have a role in regulation of vascular and neuronal regeneration after HIBI include DNA methylation, histone deacetylase, and microRNAs (miRNAs).

     DNA methylation status changes dramatically after HIBI. It could present as an increased or decreased methylation of different genes. HIBI is associated with an in-creased expression of the genes related with angiogenesis and apoptosis (Casp1, Casp9, Casp8ap2, vascular endothelial growth factor a (VEGFa), VEGFc, Epor, Epo, Hif 1α, and Hif 3α).

     The outcome of cerebral ischemia is greatly influenced by the acetylation status of histones. The creation of an ischemia-resistant state in neurons requires histone acetylation and cAMP-response element binding protein (CREB)-binding protein (CBP). The neurological outcomes are highly correlated with the expression levels of histone acetylation and CBP.

     miRNAs in the brain and blood can be used as biomarkers for cerebral ischemia. Animal studies have shown that the expression of miRNA in the hippocampus of rats is altered after global ischemic insult. miRNAs regulate the normal physiological activity in conjunction with the response to ischemia–reperfusion injury of the hippocampus. The role of miRNA in the pathogenesis of HIBI is significant, which can be explained by changes in the expression of miRNA (mir-182, mir-200b, and mir-429) after HIBI in rats. The expression of mature miR-139-5p is decreased after HIBI, which leads to increased expression of a newly identified protein, HGTD-P, and consequently promotes neuronal apoptosis in neonatal rats after HIBI. miR-210 prevents oxygen–glucose deprivation–induced apoptosis and protects cells against HIBI.

      Research into the roles of epigenetic mechanisms in cerebral ischemia is rapidly growing. However, our understanding of the epigenetics regulations for HIBI is still in its infancy. The epigenetic strategies targeting gene expression can be used for the treatment of HIBI, such as the inhibition of DNA methyltransferase activities, histone deacetylase enzyme, and miRNAs. Neuroprotective agents targeting these pathways can modulate neural cell regeneration and promote brain repair and enhance functional recovery after HIBI. A better understanding of how epigenetics influences the process and progress of cerebral ischemia will pave the way for discovering more sensitive and specific biomarkers and new targets and therapeutics for HIBI and will improve the neurological outcomes.

In the section describing pathogenesis, it would be a good idea to add timing of events occurring at the molecular-cellular level (Timeline).

Thank you. We have mentioned the details about the timing of events on page 2.

  • In clinical terms, HIBI can be attributed to two primary factors: a diffuse perfusion deficit and/or a global hypoxic insult to the brain. The timing of HIBI varies, occurring 1) before ECMO cannulation; 2) during ECMO support; or 3) during the decannulation period.

The authors were able to make a significant contribution to the development of effective clinical practices that reduce the burden of HIBI in patients with ECMO.

I recommend this manuscript for publication, after implementing the suggested improvements.

Thank you again for your valuable suggestions.

Reviewer 2 Report

The paper overviews mechanisms of developing neurological consequences and risk factors associated with the supportive extracorporeal membrane oxygenation (ECMO) treatment and focuses on the most common ECMO-induced injury, the hypoxia-ischemia (HI). In addition, the paper includes the comprehensive analysis of pathophysiology of therapeutic strategies that are employing to prevent or overcome HI post-ECMO. The advantage and novelty of the review is that it  describes the post-ECMO HI pathogenesis on both  cellular and molecular levels, with well-prepared original illustrations. The another impressive  advantage includes an enriched profile of citations directly related to the reviewed subjects  of which nearly half of sources have been published just during last 3 years (2020-2023). Not completed references 50-51 (lanes 597-599) may be easily completed for the final consummation.

Author Response

Reviewer 2

The paper overviews mechanisms of developing neurological consequences and risk factors associated with the supportive extracorporeal membrane oxygenation (ECMO) treatment and focuses on the most common ECMO-induced injury, the hypoxia-ischemia (HI). In addition, the paper includes the comprehensive analysis of pathophysiology of therapeutic strategies that are employing to prevent or overcome HI post-ECMO. The advantage and novelty of the review is that it  describes the post-ECMO HI pathogenesis on both  cellular and molecular levels, with well-prepared original illustrations. The another impressive  advantage includes an enriched profile of citations directly related to the reviewed subjects  of which nearly half of sources have been published just during last 3 years (2020-2023). Not completed references 50-51 (lanes 597-599) may be easily completed for the final consummation.

Thank you for your comment. We have updated these references.